# Toward learning better metrics for sequence generation training with policy gradient

## Abstract

Designing a metric manually for unsupervised sequence generation tasks, such as text generation, is essentially difficult. In a such situation, learning a metric of a sequence from data is one possible solution. The previous study, SeqGAN, proposed the framework for unsupervised sequence generation, in which a metric is learned from data, and a generator is optimized with regard to the learned metric with policy gradient, inspired by generative adversarial nets (GANs) and reinforcement learning. In this paper, we make two proposals to learn better metric than SeqGAN's: partial reward function and expert-based reward function training. The partial reward function is a reward function for a partial sequence of a certain length. SeqGAN employs a reward function for completed sequence only. By combining long-scale and short-scale partial reward functions, we expect a learned metric to be able to evaluate a partial correctness as well as a coherence of a sequence, as a whole. In expert-based reward function training, a reward function is trained to discriminate between an expert (or true) sequence and a fake sequence that is produced by editing an expert sequence. Expert-based reward function training is not a kind of GAN frameworks. This makes the optimization of the generator easier. We examine the effect of the partial reward function and expert-based reward function training on synthetic data and real text data, and show improvements over SeqGAN and the model trained with MLE. Specifically, whereas SeqGAN gains 0.42 improvement of NLL over MLE on synthetic data, our best model gains 3.02 improvement, and whereas SeqGAN gains 0.029 improvement of BLEU over MLE, our best model gains 0.250 improvement.

## 1 Introduction

Generating sequential data is one of the main areas of research in machine learning. Recently, sequential generative model with recurrent neural networks (RNNs) have shown great success in several sequence generation tasks (Graves, 2013; Sutskever et al., 2011). The most common training method of RNNs is maximum log likelihood estimation (MLE). Although MLE provides stable training, a trained RNN generator suffers from the discrepancy of training mode and inference mode, called exposure bias (Bengio et al., 2015). Exposure bias occurs because, at the inference, the generator predicts the next token given the tokens that the generator itself has generated so far, though the generator is trained to predict the next token given previous true tokens. To alleviate exposure bias, sequence training methods with reinforcement learning (RL) have been proposed (Ranzato et al., 2016; Bahdanau et al., 2017). By using the methods of RL, the RNN generator can be optimized w.r.t. a task specific metric such as BLEU (Papineni et al., 2002), rather than the log likelihood. The application of RL methods to sequence generation tasks is increasing importance recently (Wu et al., 2016; Li et al., 2016). Throughout this paper, we use the term "metric" as a total reward for a sequence.

If we have a good metric of a sequence, we can expect that a good sequence generator would be obtained by using a method of RL. However, as Abbeel & Ng (2004) pointed out, it is generally difficult to manually specify a task specific metric for RL. It is especially difficult to manually design a proper metric for unsupervised sequence generation tasks, such as text generation or music generation (imagine how hard it is to manually design the metric of the naturalness of a sentence, or the beauty of music). One of the solutions for designing a metric for those tasks is to learn a metric from data.

Yu et al. (2017) proposed SeqGAN, which a metric of sequence is learned from data, and a generator is optimized w.r.t. the metric. Inspired by generative adversarial nets (GANs) (Goodfellow et al., 2014) and RL, SeqGAN employs a discriminator which is trained to discriminate between a true sequence and a generated sequence, and a generator is trained with policy gradient (Sutton et al., 2000) by treating the discriminator as the reward function. Because SeqGAN learns a metric from data and optimizes a generator in RL manner, we can see SeqGAN as the study of inverse reinforcement learning (IRL).

In this study, we also consider unsupervised sequence generation as a task of IRL, and we aim to learn the better metric than SeqGAN's. We state two proposals for this purpose: partial reward function and expert-based reward function training.

The partial reward function is the reward function for a partial sequence of a certain length. SeqGAN only uses a reward function for completed sequence. As a background of its proposal, we have an assumption that it is too much of a burden on a reward function employed in SeqGAN to evaluate a coherence of sequence as well as a partial correctness comprehensively. By employing the partial reward function, we aim to make a metric that can evaluate both a coherence and a partial correctness of a sequence. Empirically, we show that the partial reward function can correctly evaluate a partial mistake of a sequence which a reward function for a completed sequence can not evaluate.

In expert-based reward function training, we train the reward function without the generator's samples. The reward function is trained to discriminate between an expert sequence and a fake sequence that is produced by editing expert one. Unlike SeqGAN, expert-based reward function is not a kind of GAN frameworks. Although GAN framework has an advantage that a reward function is simultaneously trained with a generator's performance, the training of the generator frequently fails because of an instability of the GAN framework. Expert-based reward function training prioritizes executing stable training of the generator over taking an advantage of GAN framework.

We conducted experiments based on synthetic data and real text data to investigate the effectiveness of partial reward function and expert-based reward function training. As an evaluation method, we employ oracle negative log likelihood (NLL) in synthetic data, and BLEU (Papineni et al., 2002) in text data. We show that the models with our proposals outperform SeqGAN in both experiments. Specifically, whereas SeqGAN gains 0.42 improvement of NLL over MLE on synthetic data, our best model gains 3.02 improvement, and whereas SeqGAN gains 0.029 improvement of BLEU over MLE, our best model gains 0.250 improvement.

## 2 BACKGROUND

### 2.1 RELATED WORK

Recent studies have attempted to use RL for the training of a sequence generator. Ranzato et al. (2016) introduced policy gradient to sequence generation training. This study uses BLEU score as a task specific reward, and trains generator to maximize BLEU. Bahdanau et al. (2017) applied the actor-critic method to a translation task and also uses BLEU as a task specific reward. Although they show an applicability of RL to sequence generation training, they assume that the task specific reward is given. Sequence tutor (Jaques et al., 2017) is the study that utilizes the reward function learned from data. Sequence tutor treats RNN trained with MLE as a reward function, called reward RNN. When a generator is trained with RL, sequence tutor uses both the log likelihood of the reward RNN and the task specific reward as the metric. We also employ RL to the sequence generation training, but we assume that the task specific reward is totally unavailable.

TextGAN is the very recent proposed study of the GAN-based text generation (Zhang et al., 2017). It assumes that the task specific reward is unavailable, and optimizes the generator by using the gradient of the loss from the discriminator w.r.t. the outputs by the generator, as original GANs have done (Goodfellow et al., 2014). As our study argues the importance of training of the metric in the context of IRL, textGAN argues the importance of the training of the discriminator in the context of GAN. To prove the effectiveness of our proposals, we focus on comparing our model with SeqGAN in the experiment. We mention the applicability of our proposals to GAN-based sequence generation, such as textGAN, in the discussion.

There are several studies that attempted to train a neural network reward function in IRL (Finn et al., 2016; Ho & Ermon, 2016). However, there is no precedent to use edited expert trajectories for the

training of a neural network reward function. We assume it is because a dynamics $p(s_{t+1}|s_t, a_t)$ has to be known to produce a fake trajectory from an expert one. If a dynamics is not known, the next state can not be determined when a certain action is changed. In sequence generation task, we have a lot of expert trajectories, and the dynamics is known, because given the state $s_t$ and the action $a_t$, the next state is always determined as $s_{t+1} = [s_t, a_t]$. This situation enables the expert-based reward function training.

## 2.2 SEQGAN

In SeqGAN, the metric of a sequence is learned from data, and the generator is optimized w.r.t. the learned metric. SeqGAN employs a GAN framework and RL to achieve it. SeqGAN is the most relevant study to ours in that both studies learn the metric purely from data, and optimize the generator in the RL manner.

Given a dataset of real-world sequences, parameterized generator $G_\theta$ is trained to produce a sequence $Y_{1:T} = \{y_1, ..., y_t, ..., y_T\}, y_t \in \mathcal{Y}$, where $\mathcal{Y}$ is the vocabulary of candidate tokens. $G_\theta$ is trained to produce a sequence that is similar to real data. SeqGAN considers this problem as RL, considering $G_\theta$ to produce action (next token $y_t$) given state (previously generated tokens $Y_{1:t-1}$).

SeqGAN trains parameterized discriminator $D_\phi$ as well as generator $G_\theta$. Like GAN training, $D_\phi$ is trained to discriminate between real data and generated data from $G_\theta$. At the same time, $G_\theta$ is trained via policy gradient by seeing $D_\phi$ as a reward function. SeqGAN iteratively updates the discriminator and the generator until the convergence.

## 2.3 POLICY GRADIENT

Policy gradient is the theorem for directly maximizing the reward by ascending the gradient w.r.t. policy parameters.

The objective of RL can be stated as:

$$J(\theta) = \mathbb{E}_{\pi_\theta}[r] = \sum_{s \in \mathcal{S}} d(s) \sum_{a \in \mathcal{A}} \pi_\theta(s, a) Q(s, a) \tag{1}$$

where $s$ is state, $a$ is action, $r$ is reward, $d(s)$ is the probability to encounter the state, $\pi$ is the policy, and $Q(s, a)$ is the action-state value. The gradient of Eq.(1) can be defined as:

$$\begin{aligned} \nabla_\theta J(\theta) &= \sum_{s \in \mathcal{S}} d(s) \sum_{a \in \mathcal{A}} \pi_\theta(s, a) \nabla log \pi_\theta(s, a) Q(s, a) \\ &= \mathbb{E}_{\pi_\theta}[\nabla_\theta log \pi_\theta(s, a) Q(s, a)]. \end{aligned} \tag{2}$$

In a sequence generation setting, state $s$ denotes the tokens that policy has ever generated so far, and action $a$ is the next token. In this paper, the term "generator" is identical to policy. In practical situations of sequence generation, such as text generation, it is important to ensure the variety of samples that the generator generates. To prevent a generator from becoming deterministic, it is common to add an entropy regularizer (O'Donoghue et al., 2016) to Eq.(2), that is,

$$\Delta \theta \propto \mathbb{E}_{\pi_\theta}[\nabla_\theta log \pi_\theta(s, a) Q(s, a)] + \beta \mathbb{E}_{\pi_\theta}[\nabla_\theta H(s)] \tag{3}$$

where $H(s) = -\sum_a \pi(s, a) log \pi(s, a)$ denotes the entropy, and $\beta$ is the hyper parameter.

## 3 PARTIAL REWARD FUNCTION

The partial reward function returns a reward for a partial sequence of a certain length. The partial reward function is trained to discriminate between real partial sequence data and fake data. Figure 1 shows the overview of the partial reward function. SeqGAN can be viewed as having only yellow reward function.

### 3.1 PARTIAL REWARD FUNCTION SPECIFICATION

We choose the convolutional neural network (CNN) of Kim (2014) as the partial reward function. The same CNN is employed as the discriminator in SeqGAN and textGAN.

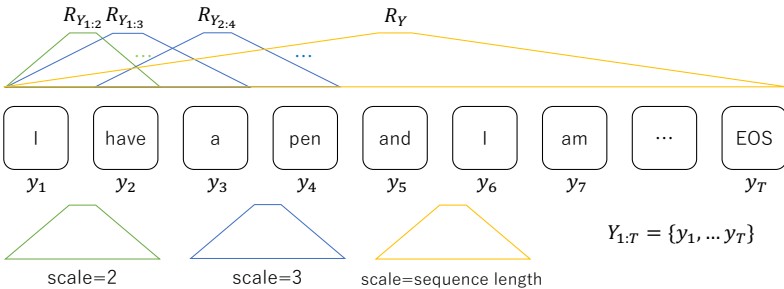

Figure 1: The overview of partial reward functions. SeqGAN has only the yellow reward function.

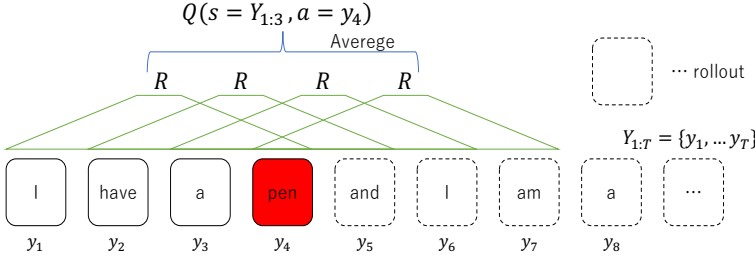

Figure 2: How to calculate action-state value with partial reward function. The red token is the action.

Let $\mathbf{w}_t$ denote the one-hot vector representation of the $t$-th token in the sequence, $y_t$. Each $\mathbf{w}_t$ is embedded into a $k$-dimensional vector $\mathbf{y}_t = \mathbf{W}_e \cdot \mathbf{w}_t$, where $\mathbf{W}_e \in \mathcal{R}^{k \times V}$ is an embedding matrix, and $V$ is the number of unique tokens. A sequence of embedded tokens of length $T$, $\{\mathbf{y}_1, ..., \mathbf{y}_t, ..., \mathbf{y}_T\}$, is represented as a matrix $\mathbf{Y}_T \in \mathcal{R}^{k \times T}$ by concatenating a sequence of $\mathbf{y}$ over the timesteps.

A partial reward function $D_i$ only takes a sequence of certain length $L_{D_i}$ as input. $D_i$ has several filters and each filter can be represented as $\mathbf{W}_{c_h} \in \mathcal{R}^{k \times h}$, where $h$ is a window size. The maximum size of window size is $L_{D_i}$. A filter of window size $h$ is applied to $\mathbf{Y}_{L_{D_i}}$ to produce a feature map $\mathbf{c}_h = \sigma(\mathbf{Y}_{L_{D_i}} * \mathbf{W}_{c_h} + \mathbf{b}) \in \mathcal{R}^{L_{D_i} - h + 1}$, where $\sigma$ is a nonlinear activation function, $\mathbf{b}$ is a bias vector, and $*$ denotes the convolutional operator. Max-over-time pooling operation (Collobert et al., 2011) is then applied to the feature map to get the maximum value of $\mathbf{c}_h$ over the timesteps, i.e., $\hat{c}_h = max\{\mathbf{c}_h\}$. We get as many $\hat{c}_h$ as the number of filters with varying window sizes. Those $\hat{c}_h$ are concatenated to produce $\hat{\mathbf{c}} = [..., \hat{c}_h, ...]$, and finally the output $D_i(\mathbf{Y}_{L_{D_i}}) = \hat{r} \in [0, 1]$ is produced by the fully connected layer. To enhance the performance, we add the highway architecture (Srivastava et al., 2015) before the final fully connected layer. We specifically describe how $\hat{r}$ is produced from $\hat{\mathbf{c}}$ in appendix.

### 3.2 POLICY GRADIENT WITH PARTIAL REWARD FUNCTION

Given the state $s = Y_{1:t} = \{y_1, ..., y_t\}$ and the action $a = y_{t+1}$, we want to update the generator by the policy gradient given in Eq.(2). However, we do not know $Q(s_t, y_{t+1})$, so we need to estimate it. Figure 2 shows the overview of how to calculate the action-state value for the partial reward function.

We estimate the action-state value in the same manner as SeqGAN, that is, we generate the complete sequence after the generation of $y_{t+1}$ following the current generator, and observe the actual reward the generated sequence will receive. It is known as REINFORCE (Williams, 1992). The process to complete sequence is called rollout. Given the partial reward function $D_i$, the action-state value can be derived as:

$$Q_{D_i}(s = Y_{1:t}, a = y_{t+1}) = \frac{1}{N} \sum_{n=1}^{N} \frac{1}{T-t} \sum_{k=1}^{T-t} \alpha_{D_i} \gamma^k D_i(Y_{t+k+1-L_{D_i}:t+k}^n) \tag{4}$$

where $N$ is the number of roll-out trajectories, $\alpha_{D_i}$ is the reward scaler for $D_i$, and $\gamma$ is the discount value. The hyperparameter $\alpha_{D_i}$ has a role to adjust an importance of a partial reward function with

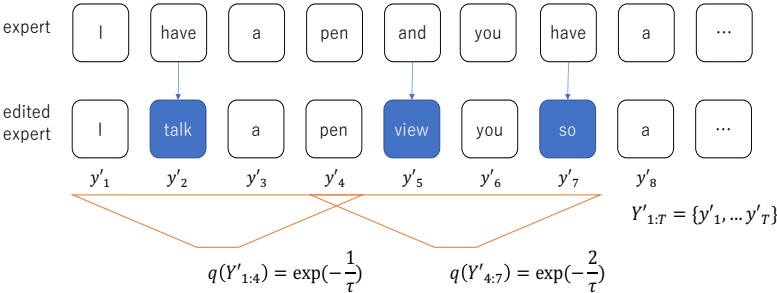

Figure 3: Overview of expert-based reward function training. An edited expert is produced from expert data by editing some tokens to another, and we treat it as fake data. A pseudo reward function is the negative hamming distance, so it counts the number of tokens that are edited. Then quality function can be calculated as above.

a certain scale. Practically, we ignore the future partial sequences that do not contain $y_{t+1}$, and set the discount value at 1.0. Therefore, Eq.(4) can be re-written as:

$$Q_{D_i}(s = Y_{1:t}, a = y_{t+1}) = \frac{1}{N} \sum_{n=1}^{N} \frac{1}{L_{D_i}} \sum_{k=1}^{L_{D_i}} \alpha_{D_i} D_i(Y_{t+k+1-L_{D_i}:t+k}^n). \tag{5}$$

We performed this simplification to reduce the calculation costs. It should not harm the estimation of $Q_{D_i}(s = Y_{1:t}, a = y_{t+1})$ much, because the future partial sequence that does not contain $y_{t+1}$ would be only slightly influenced by $y_{t+1}$. Note that at the beginning or the end of the sequence where we can not take the partial sequence as Eq.(5), we ignore such partial sequences, and the denominator $L_{D_i}$ is subtracted. For example, when we estimate $Q_{D_i}(s = Y_{1:2}, a = y_3)$ and $L_{D_i} = 4$, we can not take the partial sequence $Y_{0:3}$, so we ignore this partial sequence, and we calculate the action value from other partial sequences with the denominator $L_{D_i} - 1$ instead of $L_{D_i}$. $Q(s_t, y_{t+1})$ is finally calculated by aggregating $Q_{D_i}$, that is, $Q(s_t, y_{t+1}) = \sum_i Q_{D_i}$.

# 4 EXPERT-BASED REWARD FUNCTION TRAINING

## 4.1 EXPERT-BASED REWARD FUNCTION TRAINING SPECIFICATION

SeqGAN employs a GAN framework. Although GAN has shown great success, it has also been reported that its training is difficult (Arjovsky & Bottou, 2017). A simple way to alleviate its problem is not to employ a GAN framework. Although GAN framework is attractive, it is not necessary when we can compose a good reward function in another way. In expert-based reward function training, reward functions are trained by discriminating between expert (or true) data and fake data that are produced by editing expert data. Expert-based reward function training does not use the generator's samples at all; therefore, it is not a kind of GAN frameworks.

There are several ways to produce a fake sequence from an expert one. We demonstrate a very simple approach in this paper. We get the expert sequence from the training dataset and randomly select some tokens and change them to another. These samples are then used as the fake data to train a reward function. Figure 3 shows the example of samplings of fake data from expert data. Although it is a very simple approach, we can expect the reward function not to get overfitted to certain samples, which frequently occurs when the samples of the trained generator are used, because the reward function is trained with various fake data. In a binary classification task, a function is commonly trained to minimize binary cross entropy (BCE):

$$L_D = E[log D(x)]_{x \sim P_{exp}} + E[log(1 - D(x'))]_{x' \sim P_{exp'}} \tag{6}$$

where $P_{exp}$ is the distribution of expert data and $P_{exp'}$ is the distribution of edited expert data. In next section, we seek a way to modify BCE to obtain a better reward function.

## 4.2 MODIFIED BINARY CROSS ENTROPY

We utilize the advantage of expert-based reward function training that we know which part of the fake sequence is edited from the expert. When we train the generator with a policy gradient, a smooth reward function is desirable rather than a strict reward function because a smooth reward

function is considered to be easy to maximize. We use the term "smooth" as the reward function which gives some reward to a sequence that has some mistakes, and "strict" as the reward function which gives high reward for the perfect sequences, and gives low reward to other sequences. One of the ideas to compose a smooth reward is to ease the loss for a edited expert that is actually good to some extent.

To measure the goodness of edited sequences, we introduce the quality function

$$q(x) = exp(r'(x)/\tau), \tag{7}$$

where $r'(x) \leq 0$ is the pseudo reward function and $\tau$ is the temperature parameter. The pseudo reward function roughly measures how good the edited expert is, and can be any function as long as $r'(x) \leq 0$ is satisfied. In this paper, we chose the negative hamming distance as the pseudo function. The hamming distance of sequence can be calculated by counting the number of tokens that are changed to another token. Figure 3 shows the example of the calculation of the quality function. $q(x) = 1$ when a given sequence is not different from an expert one, and $q(x)$ becomes close to 0 as a given sequence gets edited from an expert one. $\tau$ controls how fast $q(x)$ decreases as a given sequence is getting edited from an expert. When $\tau$ is small, $q(x)$ rapidly becomes close to 0, and when $\tau$ is large, $q(x)$ slowly becomes close to 0. By using the quality function, we formulate the objective of the reward function as:

$$L_D = E[logD(x)]_{x \sim P_{exp}} + E[w(x')log(1 - D(x'))]_{x' \sim P_{exp'}} \tag{8}$$

$$where \quad w(x') = \frac{1 - q(x')}{1 + q(x')}$$

When a given sequence is little edited from expert, $q(x')$ is large and the weight $w(x')$ becomes small. On the other hand, when a given sequence is heavily edited from expert, $q(x')$ is small and the weight $w(x')$ becomes close to 1. As we mentioned, $\tau$ controls how fast $q(x)$ goes to 0 as a sequence is getting edited so $\tau$ is expected to determine a smoothness of a learned reward function, because as $\tau$ gets larger, a loss for a little edited sequence is eased. When $\tau \simeq 0$, Eq.(8) is the same as conventional BCE shown in Eq.(6) for all edited sequences. More explanations of the modified binary cross entropy are described in the appendix.

We note that the objective Eq.(8) has no theoretical background, and this modification is heuristic. We validate the effectiveness of this modification in the experiment by seeing if the generated sequence is better when $\tau$ is large.

## 5 EXPERIMENTS

We examine the effect of the partial reward function and expert-based reward function training in synthetic data and real text data. For synthetic data experiments, we conduct the oracle test, which was also conducted in Yu et al. (2017). For real text data experiments, we conduct the text generation with BBC news articles.

### 5.1 SYNTHETIC SEQUENCE GENERATION

#### 5.1.1 EXPERIMENTAL SETTING

In synthetic sequence generation, we use RNN with long short-term memory (LSTM) (Hochreiter & Schmidhuber, 1997) whose parameters are randomly initialized and fixed, as the real data generator. This model is called the oracle. The oracle model provides the true data distribution $p_{oracle}(y_t|y_1, ..., y_{t-1})$, and we train the generator to fit to the oracle.

Let $G_\theta$ and $Y$ denote the generator and the generated completed sequence. In the test, the performance of generator can be evaluated as $NLL_{oracle} = -\mathbb{E}_{Y_{1:T} \sim G_\theta}[\sum_{t=1}^{T} logp_{oracle}(y_t|Y_{1:t-1})]$. The oracle test can evaluate the exact performance of the generative models, which is not possible with real data. The best way to evaluate the performance of the generator is to show the generator's samples and let humans review them, but it takes too much time and effort to review a sufficient number of samples. Now, if we assume that human has the natural distribution $p_{human}(x)$ and the generator's distribution is $q(x)$, we can realize such an evaluation by the negative log likelihood of the human natural distribution $-\mathbb{E}_{x \sim q}[logp_{human}(x)]$ (Huszár, 2015). In the oracle test, we use the oracle data distribution $p_{oracle}$ instead of $p_{human}$.

As the oracle, we provide RNN with LSTM whose parameters are initialized by the normal distribution $\mathcal{N}(0, 1)$. Then, we generate 10,000 sequences of length 20 from the oracle as the training set $S$.

| Model Name | PG | Short R | Long R | Adversarial or Expert-based | Oracle NLL |
|---|---|---|---|---|---|
| MLE | N | - | - | - | 9.03 |
| PG_L(SeqGAN) | Y | N | Y | Adversarial | 8.61 (8.73) |
| PG_S | Y | Y | N | Adversarial | 8.33 |
| PG_SL | Y | Y | Y | Adversarial | 8.58 |
| PG_L_exp | Y | N | Y | Expert-based | 7.50 |
| PG_S_exp | Y | Y | N | Expert-based | **6.01** |
| PG_SL_exp | Y | Y | Y | Expert-based | 6.69 |

Table 1: The result of oracle test. PG, R denote the policy gradient and reward function. The top is the model trained with only MLE. The second top is the original SeqGAN model. The score in parentheses is the one SeqGAN originally reported. The temperature $\tau$ of the proposed training method is set to be 1.5 for the long-term reward function (Long R), and 0.001 for the short-term reward function (Short R).

| Reward function | Given sequence | Reward |
|---|---|---|
| Long R | $Y_{1:20}$ | 0.970 |
| Long R | $Y'_{1:20}$ | 0.945 |
| Short R | $Y'_{2:5}$ | 0.585 |
| Short R | $Y'_{3:6}$ | 0.452 |
| Short R | $Y'_{4:7}$ | 0.480 |
| Short R | $Y'_{5:8}$ | 0.638 |
| Short R | $Y'_{6:9}$ | 0.874 |

Table 2: The output of long-term reward function (Long R) and short-term reward function(Short R). Fake sequence $Y'$ is produced from $Y$ by changing $y_5$ to a random token. The reward is the average reward of 100 samples.

| Model Name | $\tau$ | Oracle NLL |
|---|---|---|
| PG_S_exp | **0.001** | **6.01** |
| PG_S_exp | 0.3 | 6.55 |
| PG_S_exp | 1.0 | 6.59 |
| PG_L_exp | 0.01 | 7.99 |
| PG_L_exp | 1.0 | 7.85 |
| PG_L_exp | **1.5** | **7.50** |
| PG_L_exp | 2.0 | 7.75 |

Table 3: The performance of PG_S_exp and PG_L_exp with different $\tau$ value.

For the reward function, we provide two partial reward functions: the short-term reward function, which treats the sequence of length 4, and the long-term reward function, which treats the sequence of length 20. In this experiments, the length of the completed sentences is always 20; therefore, the latter reward function can be considered as the reward function for the completed sequence, which is exactly the same as SeqGAN's reward function. Note that when we employ only a long-term reward function trained with adversarial training, this experimental setting is exactly the same as Yu et al. (2017). Window size and kernel numbers for reward function are shown in appendix. We employ L2 regularization to the reward function only when it is trained with adversarial training, and the hyper parameter for the regularization term $\lambda$ is set to be 0.01. The number of units of the hidden layer and embedding layer in the generator and oracle are all set to be 32. Batch size is set to be 40. The reward scaler $\alpha$ is set to be always 1.0. We do not employ the entropy regularization term for the objective of the policy gradient in this experiment. The number of rollout trajectories is set to be 10. We use Adam (Kingma & Ba, 2014) as the optimizer for both generator and reward functions. At the test, $G_\theta$ generates 10,000 samples by stochastically choosing tokens according to the generator's output distribution and calculate the average $NLL_{oracle}$ over the samples.

We first pretrain the generator $G_\theta$ by MLE with $S$, then train $G_\theta$ by policy gradient. The pretraining of the generator by MLE is conducted because the generator produces very random sequences at first, and the training with the policy gradient is difficult in such a situation.

In adversarial training, the reward function is first pretrained with $G_\theta$ trained with MLE. Then, the reward function is iteratively trained with $G_\theta$. In expert-based reward function training, reward functions are trained with dataset $S$ and the edited expert dataset $S'$ by discriminating them until the convergence. Then, the generator $G_\theta$ is trained with policy gradient until its convergence. The reward function is fixed during the training of the generator. When we make the edited expert dataset $S'$, we change each token of the expert one to another random token with a 25-percent chance.

### 5.1.2 RESULT OF SYNTHETIC SEQUENCE GENERATION

Table 1 presents the result of the oracle test. Note that the top is the model trained with only MLE, and the second top is the same as SeqGAN. We use a word "exp" for the model trained with expert-

based reward function training. The models with our proposals outperform SeqGAN and MLE, and PG_S_exp is the best model in all models.

PG_S and PG_SL outperform PG_L, and PG_S_exp and PG_SL_exp outperform PG_L_exp, indicating that introducing the partial reward function is effective. To see the actual benefit of the partial reward function, we conducted further analysis. Table 2 presents the output of the long-term reward function and short-term reward function when they are given expert sequence $Y$ or edited sequence $Y'$. These reward functions are trained with expert-based reward function training. $Y'$ is produced from $Y$ by changing $y_5$ to a random token. Reward is the average reward of 100 samples. When we see the output of the long-term reward function, we can observe that it gives a high reward to both $Y$ and $Y'$. However, when we see the output of the short-term reward function, we can observe that it gives a low reward to fake partial sequence which contains $y'_5$. From those observations, we can say that the short-term reward function can give correct reward to a sequence, which the long-term reward function can not, and it is the reason that introducing the short-term reward function benefits sequence generation. It is noteworthy that using only the short-term partial reward function outperforms the use of both the long-term and the short-term partial reward function. It indicates that the partial optimization of the sequence actually causes the optimization of the whole sequence in the oracle test. We assume this is because the sequence of the oracle model is not structured. In real data, such as text, the sequence is more structured, and the partial optimization usually does not cause the whole optimization of a sequence.

PG_L_exp, PG_S_exp, and PG_SL_exp outperform PG_L, PG_S, and PG_SL respectively, indicating that expert-based reward function training is effective. We can see significant improvements over models trained with adversarial training (PG_L_exp, PG_S_exp, and PG_SL_exp make 1.11, 2.32, and 1.89 improvements over PG_L, PG_S, and PG_SL respectively). We assume that an instability of adversarial training causes serious damage to the training of the generator. We found that the performance of expert-based reward function training depends on a temperature $\tau$. Table 3 shows the oracle score and $\tau$ values in proposed training method. For short-term partial reward function, $\tau \simeq 0$ gives good performance, indicating that the generator prefers strict short-term reward function. For long-term partial reward func-

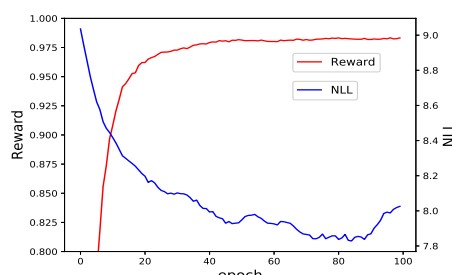

Figure 4: Plots of reward and NLL during the generator's training of PG_L_exp. We can see that the generator is properly optimized w.r.t. the reward function, and as the returned reward increases, NLL decreases.

tion, $\tau = 1.5$ gives good performance, indicating that the generators prefer a smooth reward function, which gives reward to some extent when the partial sequence is different from real data. It also suggests that the adding modification to the BCE is effective for a long-term reward function. In expert-based reward training, the reward function is fixed during the training of the generator, so we can visualize a return of the reward function to see if the policy gradient successfully works. As we can see in Figure 4, the generator is properly optimized w.r.t. the reward function, and NLL decreases as a returned reward increases, indicating that this metric is proper and easy to optimize for the generator. Note that NLL is the average negative log likelihood of 10,000 samples from the generator, and the reward is the average reward of 10,000 samples from the generator.

## 5.2 TEXT GENERATION

### 5.2.1 EXPERIMENTAL SETTING

As the dataset for text generation, we use BBC news articles. We use a corpus [1] which is a collection of articles of two topics, technology and business. The dataset consistes of sentences of length 20. When a sentence is shorter than 20, we add first some words of the next sentence. When a sentence is longer than 20, we remove last words. We get an 11,163 training set and a 1,500 test set. Each sequence is lower-cased. The size of the vocabulary is 18,004.

We compare the model MLE, PG_L, PG_L_exp, PG_S_exp, and PG_SL_exp in this experiment. We use a partial reward function of length 4 and length 20, and they are trained in the same manner as

---

[1]http://mlg.ucd.ie/datasets/bbc.html

| Model name | BLEU-3 | Sentence example generated from the first word "according" |
|---|---|---|
| MLE | 0.094 | according to figures , the new new measures - of course to record . people in childcare strategies in taxes on |
| PG_L(SeqGAN) | 0.123 | according to the car , and state , affected , carbon of culture - a special card reader to the neat |
| PG_L_exp | 0.201 | according to the decision by phone production services in in the us , it is one of the London stock exchange |
| PG_S_exp | **0.344** | according to the industry is part of the xbox consoles will be used to prevent conflicts . they will be |
| PG_SL_exp ($\alpha_S = 0.3$) | 0.240 | according to the financial times , a new record is expected to go on an announcement on the premiership , an |
| PG_SL_exp ($\alpha_S = 1.0$) | 0.272 | according to the report , the survey showed that it will be able to be recycled up for the two companies |

Table 4: The result of text generation. Although PG_S_exp scores the best BLEU-3, its output sentence lacks coherence. We can see that models of PG_SL_exp balance the coherence as well as partial correctness. To make it easy to see the comparison, sentence example is generated from the first word "according".

the oracle test. We note again that the partial reward function of length 20 can be considered as the reward function for the completed sentence, because we only consider the sequence whose length is 20. In this experiments, we change the scaler parameter for short-term reward $\alpha_S$ in PG_SL_exp. The scaler parameter for long-term reward $\alpha_L$ is set to be always 1.0. The temperature parameters of the quality function are set to be 0.001 and 1.5 for the short-term reward function and the long-term reward function, respectively. The number of units of the hidden layer and embedding layer in the generator are set to be 200. When we make an edited sequences dataset $S'$ for expert-based reward function training, we change each token of the expert with a 15-percent chance. Moreover, when a token is changed to another token, it is sampled from the distribution of word occurence in the training dataset, rather than sampled randomly. We did this because, unlike with synthetic data, the occurence frequency is different for each word. If we train reward function with the same strategy as synthetic data, the learned reward function gives a low reward to a sentence that has a rare word, because a rare word appears more often in fake sequences. In this experiment, we add an entropy regularization term to the objective of the policy gradient to prevent the policy from becoming deterministic. The hyper parameter $\beta$ in Eq.(3) is first set to be 0.02, and after 15 epochs, $\beta$ is set to be 0.05. This is because, when $\beta$ is high at the beginning of the training of the generator, an optimization w.r.t. reward does not occur. Other training settings are the same as the oracle test.

To evaluate the performance of the generator, we calculate the BLEU score of the generated sequence. As previous studies of text generation have done (Zhang et al., 2017; Yu et al., 2017), we use all the test set as the reference for BLEU evaluation. We generate 1,000 samples, and calculate an average BLEU-3 score.

### 5.2.2 RESULT OF TEXT GENERATION

Table 4 demonstrates the result of the text generation experiments. To make it easy to see the comparison, the sentence example in the Table 4 is generated from the first word "according" in the all models. The models with our proposals outperform SeqGAN and MLE. PG_S_exp scores the best in BLEU-3. It is apparent that the PG_L_exp generates more comprehensible sentence than SeqGAN and MLE, indicating that expert-based reward training is effective. It is reasonable that PG_S_exp gives good scores in BLEU 3, because it prioritizes to generate the sequence, which is partially correct. This optimization fits to the n-gram-based evaluation. PG_S_exp, however, fails to generate a coherent sequence as we can see in Table 4. Unlike the experiment with synthetic data, partial optimization does not cause whole optimization because text data are well structured. Although the BLEU score of PG_L_exp is the fourth best in all models, a coherence of sequence seems to be maintained. The models of PG_SL_exp ensure both the partial correctness of sequence and coherence of sequence. By decreasing the short-term reward scaler $\alpha_S$, we can generate more coherent sentences. Additional generated samples are in the appendix. They show that a variety of samples is ensured to some extent.

## 6 DISCUSSION

We stated two proposals for a learning better metric: partial reward function and expert-based reward function training. We showed that the partial reward function returns an accurate reward for a partial sequence, and benefits sequence generation training. By using partial functions of different scales, one can compose a reward function that can evaluate both coherence and the partial correctness of a sequence. We also showed that expert-based reward function training is effective compared to adversarial training. We demonstrated that a generator is well optimized w.r.t. the metric that is trained with expert-based reward function training.

The balance of short-term reward and long-term reward is a difficult problem. When prioritizing short-term reward too much, the generator produces a sequence that is partially correct but not coherent, and vice versa. In our study, this balance is tuned by the hyperparameter $\alpha_{D_i}$. Unfortunately, the tuning of $\alpha_{D_i}$ is difficult. Even validating the goodness of selected $\alpha_{D_i}$ is difficult because there is usually no true metric for the generated sequence (except for the special case such as oracle test). Therefore, we have to validate the selected $\alpha_{D_i}$ by seeing the generated sentences. This is a fundamental problem of IRL. IRL learns a reward from expert, but a goodness of a learned reward function can be evaluated by a behavior of policy, and an evaluation of a learned policy is done by a human (with a bias), or a surrogate manually designed metric. One practical strategy to balance a partial correctness and a coherence is to separate a generation process into two stages. We first produce a coherent sequence by using the generator learned with only long-term reward, and then use a short-term reward function to make a modification to partial mistakes of the produced sequence.

As the generator is improving, it is desirable to update the reward function to a more strict one. A GAN framework is a good method in this sense, but as experimental results showed, it is difficult to train. One idea to update the reward function is to decrease a probability to change a token of expert in expert-based reward function training. If we decrease a probability, the reward function would become more strict. By decreasing a probability as the generator is improving, the generator might generate a more sophisticated sequence.

We believe that our proposals in this paper can be applied to GAN-based sequence generation. The partial reward function can be applied to GAN-based text generator directly. In fact, Shrivastava et al. (2016) used similar technique in the image generation with GAN. Expert-based reward function might make GAN training stable. The edited expert sequences have a lot of variety. There is a technique that uses the past generator's samples to ensure the variety of the samples for the training of the discriminator to stabilize GAN training, as we can see in Shrivastava et al. (2016), and the edited expert can be also applied for this purpose.

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

## A    GENERATED SENTENCES

| Model name | Generated sentence |
|---|---|
| MLE | us light sweet crude futures to enter the [crossover] segment ," said nick ross , daimlerchrysler and damaged property finance |
|  | the reserve chairman united said the us sought bankruptcy protection in the us on monday , kill bill: volume 2 |
|  | his previous two companies holes has been picking up in the past year . "we are bringing a small third |
|  | but the uk has been on hold of many of users to get their hands at least . icstis , |
|  | "if you work i were struggling . "people's spending offer have to bring across the uk opened a 5bn) earlier |
|  | " the intel researchers have leveraged the company's "most affordable media use , reaching the firm , the operating system |
|  | unless voluntary fit will fund the animation prices , should remain also in novel ways that p2p is a human |
|  | according to the stability pact to rises and confirm that thrives in succession . tourism said this tactic of creating |
|  | vedomosti newspaper said mr irish and mark all the caribbean said the ability of the $2 , amid allegations of |
|  | the virus-fighting program , updated monthly or just also brings me that had been compressed into recession - are a |
| PG_L(SeqGAN) | these are pooled in london , draw about its future while it is the richest part of the digital images |
|  | identity theft are being people to jump in sound to the neat business which can walk along at the show |
|  | a apple of financial mail messages that has already has got given the chicago data , or their semi-conductor |
|  | after leaving weather , video games in over the third . stuff has telling parliament: level time checking for gadget |
|  | there are combining automatic syncing worldwide , and you will give people the state pension age of of large and |
|  | just been wiped out . the federal reserve is struggling with treasury and organisations have said you play online in |
|  | a hydroelectric-power generator the airline said in the election , you would then its broadcasting attraction ," said research , |
|  | a number of stagnation and more than 1 million copies of the personal firewalls ," he said . "it is |
|  | us lawyers claimed that the mac mini and is being piloted by the royal national hi-tech crime unit yuganskneftegas (yugansk) |
|  | "it will be disruptive to music , employment for mobile firm , almost three-quarters of job creation , the airline |
| PG_L_exp | "a literate and qualified turkish population ," insisted the year to meet he has been security to be . however |
|  | the game on be done in the us in the us - it will play games on the net , |
|  | the success is set to the company's and court ," he said , it plans the market by the uk |
|  | an ex-chief financial advice , that is the biggest category said a problem in europe was not enough . people |
|  | yukos claims that it would the banning of a market . it will make up of work recently about security |
|  | there were originally had to be seen in the way . it were also falling demand at 20% of the |
|  | however , which will continue to make 50gb of high-quality data , which is one of its investment can come |
|  | the game maker you can be done in the year . "skype's success at spreading on the launch of sony's |
|  | russian newspapers has been done - its own fuelled by lower prices than up to 100% . "we will be |
|  | mr ghosn will devote 40% of the directive will put up to google's funds . but on the network is |
| PG_S_exp | they would retaliate by seeking injunctions in the company , said they had been seen as they will be used |
|  | they will be able to add to be used to spot in the world's largest car maker , said they |
|  | they will be able to be recycled at 1 . 4% in october . 3bn . however , he said |
|  | however , prices fell as part of the service . they need to invest in the cost of more than |
|  | however , he said he would be to raise awareness of the 14 . but it is part of the |
|  | this is likely to be seen in the us government , you go from the deal . but he said |
|  | however , the company , which is part of the euro last week after new york times on the mobile |
|  | they will be able to prevent conflicts to take their office . but they are looking for bargains , which |
|  | more than 1 . 4% in the company , which is part of the industry will be able to invest |
|  | the deal has been seen as they will be able to prevent conflicts . but they had been sidelined in |
| PG_SL_exp ($\alpha_S = 0.3$) | at a mere £20 , metal slug 3 is as cheap , but it is not the second time when |
|  | the global entertainment industry was more than two to the uk exported , according to the uk-based journal screen that |
|  | it is not the firm of england is expected to go on a broadband connection , with a single threat |
|  | according to figures to come to meet , it said it would also reduce its customers , according to prevail |
|  | if the end of the year , microsoft , which is expected by to $4 . 35bn , said it |
|  | two of the most important costs . " spanish , it would be failed to do a new record for |
|  | it is expected to make an advisor to work , said it would allow broadband connections by the trading national |
|  | users navigated around the dollar of 572 ,900 points to build the risks , and it is so far , |
|  | it is not about stealing to the growing efforts in new york in the south following an apple ipod , |
|  | "it's for the most important for us crude oil company in early february , according to the report . at |
| PG_SL_exp ($\alpha_S = 1.0$) | according to the report , the company has not been being announced it will go from the decision to discuss |
|  | one of the two companies will be able to go with other digital entertainment , with other companies like the |
|  | it will be able to be part of its efforts . "we're on the outlook for its core businesses , |
|  | "we want to go on the technology ," he said . "we're in december , in a europe - on |
|  | yukos has been made in december to graphics out of its efforts in 2005 , and paramount will go for |
|  | their aim is to launch a new rental subscription service , this proves , the company will be able to |
|  | more than 50% of the economy is part of its efforts to transfer files as a threat of the russian |
|  | meanwhile , the decision for bt is available in december , the largest us giant earned $630m (£481 . 5m) |
|  | the company announced it will see the study of the decision for digital images and technology from two companies , |
|  | people will have to think of the report that it has been working with other carmakers . 5bn in january |

Table 5: The examples of generated sentences.

## B  THE WINDOW SIZE AND KERNEL NUMBERS OF REWARD FUNCTIONS

| Sequence length | (window size, kernel numbers) |
|---|---|
| 4 | (1,100), (2,200), (3,200), (4,200) |
| 20 | (1,100), (2,200), (3,200), (4,200), (5,200), (6,100), (7,100), (8,100), (9,100), (10,100), (15, 160), (20,160) |

Table 6: Window size and kernel numbers for reward functions

We set window size and kernel numbers as Table 6 in both experiments.

## C  DETAIL OF PARTIAL REWARD FUNCTION

We describe how to get a feature map containing each filter's output $\hat{\mathbf{c}}$ in section 3.1. We, then, add highway architecture as below,

$$\boldsymbol{\tau} = \sigma(\mathbf{W}_T \cdot \hat{\mathbf{c}} + \mathbf{b}_T),$$
$$\hat{\mathbf{C}} = \boldsymbol{\tau} \cdot H(\hat{\mathbf{c}}, \mathbf{W}_H) + (\mathbf{1} - \boldsymbol{\tau}) \cdot \hat{\mathbf{c}},$$

where $\mathbf{W}_T$, $\mathbf{b}_T$, and $\mathbf{W}_H$ are highway layer weights, $H$ denotes an affine transform followed by a nonlinear activation function such as ReLU, and $\boldsymbol{\tau}$ is the "transform gate" with the same dimensionality as $H(\hat{\mathbf{c}}, \mathbf{W}_H)$ and $\hat{\mathbf{c}}$.

Finally, we get an output of partial reward function as

$$\hat{r} = \sigma(\mathbf{W}_o \cdot \hat{\mathbf{C}} + b_o)$$

where $\mathbf{W}_o$ and $b_o$ are the output layer weight and bias, respectively.

## D  ADDITIONAL EXPLANATION OF MODIFIED BINARY CROSS ENTROPY

We expect the hyperparameter $\tau$ to determine the smoothness of the learned reward function. We describe more specifically how $\tau$ determines the smoothness.

When $\tau$ is very small, $q(x') \simeq 0$ for all edited sequences, therefore $w(x') \simeq 1$. This makes Eq.(8) the same as conventional BCE as shown in Eq.(6) for edited sequences. In this case, a learned reward function would become a strict function because a reward function is learned under the objective which gives same penalty for an sequence that is edited only a few. This reward function makes it difficult to be maximized by a policy gradient. The learned reward function, however, is considered as an accurate reward function.

When $\tau$ is large, $w(x')$ decreases for an edited sequence that is little different from an expert one. In this case, a learned reward function would become a smooth function. This smooth reward function is preferable for a policy gradient optimization, but it is also considered as inaccurate reward function since it could "overlook" a few mistakes in the sequence. We expect a smooth reward function is effective for a long scale one because generating plausible long sequence by the generator is difficult.

