# OpenReview forum: "Toward learning better metrics for sequence generation training with policy gradient"
_ICLR.cc/2018/Conference — Reject_

### Official Review · AnonReviewer1 · 2017-11-21
**Using IRL techniques instead of GANs for sequence generation**

**Rating:** 7
**Confidence:** 1

**Review:**

This article is a follow-up from recent publications (especially the one on "seqGAN" by Yu et al. @ AAAI 2017) which tends to assimilate Generative Adversarial Networks as an Inverse Reinforcement Learning task in order to obtain a better stability.
The adversarial learning is replaced here by a combination of policy gradient and a learned reward function.

If we except the introduction which is tainted with a few typos and English mistakes, the paper is clear and well written. The experiments made on both synthetic and real text data seems solid.
Being not expert in GANs I found it pleasant to read and instructive.

---

> ### Author Response · Authors · 2017-12-04
> **Reply**
>
> Thank you for the review. I am glad that you enjoyed reading our paper.
> About the mistakes of English in the introduction part, we will get native check and revise it.

---

### Official Review · AnonReviewer2 · 2017-11-24
**The paper introduces an RL approach to generating time series data without the difficult training of GANs. Unfortunately, the paper is too poorly written to be clear or effective.**

**Rating:** 4
**Confidence:** 3

**Review:**

This paper describes an approach to generating time sequences by learning state-action values, where the state is the sequence generated so far, and the action is the choice of the next value.  Local and global reward functions are learned from existing data sequences and then the Q-function learned from a policy gradient.

Unfortunately, this description is a little vague, because the paper's details are quite difficult to understand.  Though the approach is interesting, and the experiments are promising, important explanation is missing or muddled.  Perhaps most confusing is the loss function in equation 7, which is quite inadequately explained.

This paper could be interesting, but substantial editing is needed before it is sufficient for publication.

---

> ### Author Response · Authors · 2017-12-04
> **Reply**
>
> Thanks for the review.
>
> From the title and the first paragraph of your review, we assume that you might not get our paper, maybe due to our poor writing. We are not sure how you understand our paper, so we firstly try to correct your misunderstandings.
>
> This paper is introducing the two techniques to learn better reward function, partial reward function and expert-based reward function training, rather than introducing new RL approach. From your review, it can be assumed that you think our paper argues about q-learning, but our paper uses policy-based RL approach (it has been firstly done by Ranzato et al. and it is not our novelty) and does not argue about q-learning at all. A policy (or a sequence generator) is learned by a policy gradient, and Q-function is NOT learned by a policy gradient. In REINFORCE, Q-value is estimated by Monte-Carlo samplings. I think the first paragraph of reviewer3 well summarizes our paper. We would appreciate if you could tell us which parts of our paper actually caused your misunderstandings so that we can revise these parts.
>
> Q. Explain about equation 7 specifically.
> A. The motivation of equation 7 is, when the produced fake sequence is not quite different from the true sequence (for example, only one token in the sequence of length 20 is changed), we thought it would be effective to decrease the weight of the objective function, binary cross entropy (BCE), because this fake sequence is actually not so bad sequence. The benefit of decreasing the weight for such sequence is that the learned reward function would become easier to be maximized by a policy gradient, because learned reward function would return some reward to a generated sequence that has some mistakes. In our paper, we describe it as “smooth" reward function.
> The parameter \tau in quality function directly affects the weight of BCE. When \tau is large, the fake sequence that is little edited from expert one get a large value of quality function, resulting in making (1 - q) / (1 + q) lower than 1, and it decreases the weight of the second term in the right hand side of equation (7). On the other hand, when \tau is small, the fake sequence that is little edited from expert one gets a near 0 value of quality function, resulting in (1 - q) / (1 + q) ~= 1, and equation (7) becomes the conventional BCE.
> The term (1 - q) / (1 + q) is heuristic and there is no theoretical background for it, but it enables to control the strictness of the learned reward function by changing the parameter \tau (“strict” means that only realistic sequence gets the reward close to 1, and others get the reward close to 0. A strict reward function is accurate, but it is considered to be difficult to maximize by a policy gradient because this reward function might be binary-like peaky function). In the experiment, we show that when the partial reward function has long scale, easing the conventional BCE by using \tau=1.5 is effective.
>
> Please give us more specific parts that you are still confused, and we are willing to give answers.
>
> Best,

---

> > ### Comment · AnonReviewer2 · 2017-12-12
> > **Good response, but still not ready for publication**
> >
> > Given the thorough response and the other reviews, I went back to re-read the paper to make sure I was being fair.  I was a little harsh, but still don't believe this paper is ready for publication, as important paragraphs are quite difficult to read and parse.  I have changed my review from a 3 to a 4.
> >
> > As an example of points that are unclear:
> >
> > 2.1: it's quite unclear what you mean by "dynamics" at the end of this section which are known in the sequence generation task, confusing this explanation.
> > 3.1: W_e isn't mentioned again, making it unclear what space you're learning in.
> > 3.2: selection of alpha_D_i isn't discussed, though discounted by the fact I haven't looked at REINFORCE in some time.  It seems it would matter quite a lot.
> > 4: Your discussion above on equation 7 helps a lot, and would benefit the paper (though I still wouldn't quite advocate acceptance).  This is particularly true since elements are "heuristic," as you say, making it non-obvious where they came from.  This is perhaps the core of my concerns with this paper: crucial equations we are to take on faith, without justification or explanation, should not be published.  It is very confusing to try and re-derive equation 7 from the points made in the preceding parts of the paper; it just doesn't follow without much more explanation.

---

> > > ### Author Response · Authors · 2017-12-16
> > > **Reply**
> > >
> > > Thanks for the reply and giving the specific parts of the paper that are unclear.
> > > We are giving answer to these questions.
> > > Moreover, we revised our paper to satisfy your request.
> > >
> > > Q, What does “dynamics” mean?
> > > A. This is where our explanation lacks. I give more specific explanation.
> > > “dynamics” means the transition probability of the next state given the current state and action, formally p(s_{t+1} | s_{t}, a_{t}).
> > > In a lot of tasks in reinforcement learning, dynamics is usually unknown and difficult to learn.
> > > In a sequence generation, however,  s_{t} is the sequence that the generator has generated so far and a_{t} is the next token generation, and s_{t+1} is always [s_{t}, a_{t}], therefore p(s_{t+1} | s_{t}, a_{t}) is deterministic. So, the dynamics is known.
> > > This nature is important when we generate fake sequence from expert, like our method. If we do not know the dynamics, we can not determine the next state when we change the certain action.
> > >
> > > We revised the section 2.1 by adding those explanation.
> > >
> > > Q,W_e isn't mentioned again, making it unclear what space you're learning in.
> > > A. W_e is just the embedding matrix (it is learned together with other weights) and we specified the dimension of embedding layer in the description of the experiment section (In synthetic data, the dimension of embedding layer is 32, and in text data, it is 200).
> > > Does it answer your question?
> > >
> > > Q, The selection of \alpha.
> > > A. The selection of \alpha is important when we use partial reward functions of different scales, because it balances the priorities of the partial correctness of different scale length. Our paper probably should argue it more specifically.
> > >
> > > Unfortunately, the selection of  \alpha_{D_i} is done by nothing but hyper-parameter tuning, and we are aware that it is the problem as we argued in the discussion section. In the text generation task, we prepare two partial reward functions (Long R and Short R), and empirically show the differences of BLEU score and generated sequence when \alpha is changed. The fact that a true metric for sequence is usually not given (except for the special case, such as oracle test) makes difficult to even validate the goodness of selected \alpha_{D_i}. This is the reason we only try \alpha_s = 0.3. and \alpha_s = 1.0 in the text generation experiment.
> > >
> > > I think this problem is not only in our case, but the fundamental problem of inverse reinforcement learning (IRL). IRL learns a reward from expert, but the goodness of learned reward function can be evaluated by the behavior of policy, and the evaluation is done by a human (with a bias), or a surrogate manually designed metric.
> > >
> > > Above discussion is included in the discussion (and a little explanation is added in 3.2).
> > >
> > > Q, Some concerns about equation 7.
> > > A. We understand your main concerns.
> > > In our paper, equation 7 comes from nowhere, and we do not clearly say that it is completely heuristics. This would confuse readers as you were so.
> > >
> > > We, however, believe that even though the justification of equation 7 is not done in a theoretical way, the justification can also be done in an experimental way. If there is a proper experimental validation for a proposal, the proposal should be the important contribution to the community.
> > >
> > > We revised our paper as below to make section 4 clear.
> > > We divided the section 4 into the two subsections 4.1 and 4.2, the one for proposing the idea of expert-based reward function training, and the other one for proposing the modified objective function.
> > > In the second subsection, we clearly wrote that
> > >
> > > - objective function comes by heuristics and there is no theoretical justification.
> > > - when \tau ~= 0, this objective function becomes conventional binary cross entropy.
> > > - The effectivity of this objective function is validated in the experiment section.
> > >
> > > and more specific explanation for the objective as we discussed in the reply for your first review.
> > >
> > > Please have a look at the revised version and give us a reply if you have any other concerns.
> > >
> > > Best,

---

### Official Review · AnonReviewer3 · 2017-11-30
**A novel contribution to sequence generation**

**Rating:** 7
**Confidence:** 3

**Review:**

This paper considers the problem of improving sequence generation by learning better metrics. Specifically, it focuses on addressing the exposure bias problem, where traditional methods such as SeqGAN uses GAN framework and reinforcement learning. Different from these work, this paper does not use GAN framework. Instead, it proposed an expert-based reward function training, which trains the reward function (the discriminator) from data that are generated by randomly modifying parts of the expert trajectories. Furthermore, it also introduces partial reward function that measures the quality of the subsequences of different lengths in the generated data. This is similar to the idea of hierarchical RL, which divide the problem into potential subtasks, which could alleviate the difficulty of reinforcement learning from sparse rewards. The idea of the paper is novel. However, there are a few points to be clarified.

In Section 3.2 and in (4) and (5), the authors explains how the action value Q_{D_i} is modeled and estimated for the partial reward function D_i of length L_{D_i}. But the authors do not explain how the rewards (or action value functions) of different lengths are aggregated together to update the model using policy gradient. Is it a simple sum of all of them?

It is not clear why the future subsequences that do not contain y_{t+1} are ignored for estimating the action value function Q in (4) and (5). The authors stated that it is for reducing the computation complexity. But it is not clear why specifically dropping the sequences that do not contain y_{t+1}. Please clarify more on this point.

---

> ### Author Response · Authors · 2017-12-04
> **Reply**
>
> Thanks for the review.
> Your first paragraph of the review well summarizes our paper. Our paper is seemingly well understood by you.
>
> Q. How are the action-state values of different length aggregated?
> A. We simply add the Q values of different scales. To balance the importance of different scales, we also introduce hyper parameter alpha.
>
> Q. Why are the future subsequences that do not contain y_{t+1} ignored?
> A2. In some setting such as Go or Atari games, the final state of the agent is important (e.g. win or lose), and future states affect the Q-value a lot. So, it is important to see further future state after the certain action at t to estimate Q-value in those setting. In our setting, however, the importance of states (or subsequences) does not depend on the timesteps. The partial reward functions treat every subsequences at a time step equally. So, we think the subsequences that contain y_{t+1} are enough samples (and they should depend on q-value of y_{t+1} a lot because y_{t_1} itself is in the subsequences) to estimate q-value.
> In equation (4), the subsequences that do not contain y_{t+1} are not ignored.

---

### Author Response · Authors · 2017-12-16
**We revised the paper**

Given the valuable reviews, we revised the following parts of our paper.

2.1 We add the description of why the dynamics is known in the sequence generation setting.

3.2 We add the description of the \alpha_{D_i} that it adjusts the importance of a partial reward function with a certain length.

3.2 We describe that Q is finally calculated by aggregating all Q_{D_i}.

4    We divide this section into two, because 4 has two contents, the proposal of expert-based reward function, and the modification of the objective. By receiving the comment from reviewer2, we wrote that the modified BCE has no theoretical background and is a heuristic. The justification of this objective is done by experimental way.

5.1.2 We state that PG_L_exp gets benefit when \tau=1.5, indicating that the modified BCE is effective.

6   We discuss the selection of \alpha_D and its difficulty.

---

> ### Author Response · Authors · 2018-01-05
> **Further revision**
>
> 1. We fixed some typos and grammar mistakes
>
> 4.1 The title of section is substituted to "expert-based reward function training specification" because previous seciton title does not suit
>
> 4.2 moved some explanation of modified binary cross entropy to appendix because it was bit verbose
>
> 5.2.2 We changed the generated examples in Table 4 to make it easy to see the comparison. All generated examples are started from the word "according".

---

### Decision · Program_Chairs · 2018-01-29
**ICLR 2018 Conference Acceptance Decision**

**Decision:**

Reject

**Comment:**

The pros and cons of this paper can be summarized as follows:

Pros:
* It seems that the method has very good intuitions: consideration of partial rewards, estimation of rewards from modified sequences, etc.

Cons:
* The writing of the paper is scattered and not very well structured, which makes it difficult to follow exactly what the method is doing. If I were to give advice, I would flip the order of the sections to 4, 3, 2 (first describe the overall method, then describe the method for partial rewards, and finally describe the relationship with SeqGAN)
* It is strange that the proposed method does not consider subsequences that do not contain y_{t+1}. This seems to go contrary to the idea of using RL or similar methods to optimize the global coherence of the generated sequence.
* For some of the key elements of the paper, there are similar (widely used) methods that are not cited, and it is a bit difficult  to understand the relationship between them:
** Partial rewards: this is similar to "reward shaping" which is widely used in RL, for example in the actor-critic method of Bahdanau et al.
** Making modifications of the reference into a modified reference: this is done in, for example, the scheduled sampling method of Bengio et al.
** Weighting modifications by their reward: A similar idea is presented in "Reward Augmented Maximum Likelihood for Neural Structured Prediction" by Norouzi et al.

The approach in this paper is potentially promising, as it definitely contains a lot of promising insights, but the clarity issues and fact that many of the key insights already exist in other approaches to which no empirical analysis is provided makes the contribution of the paper at the current time feel a bit weak. I am not recommending for acceptance at this time, but would certainly encourage the authors to do clean up the exposition, perhaps add a comparison to other methods such as RL with reward shaping, scheduled sampling, and RAML, and re-submit to another venue.